# Emerging Perspectives on Pain Management by Modulation of TRP Channels and ANO1

**DOI:** 10.3390/ijms20143411

**Published:** 2019-07-11

**Authors:** Yasunori Takayama, Sandra Derouiche, Kenta Maruyama, Makoto Tominaga

**Affiliations:** 1Department of Physiology, Showa University School of Medicine, 1-5-8 Hatanodai, Shinagawa, Tokyo 142-8555, Japan; 2Thermal Biology group, Exploratory Research Center on Life and Living Systems, National Institutes for Natural Sciences, 5-1 Aza-higashiyama, Myodaiji, Okazaki, Aichi 444-8787, Japan; 3National Institute for Physiological Sciences, National Institutes for Natural Sciences, 5-1 Aza-higashiyama, Myodaiji, Okazaki, Aichi 444-8787, Japan

**Keywords:** TRPA1, TRPV1, TRPM3, ANO1, acute pain, inflammatory pain, migraine, Candidiasis

## Abstract

Receptor-type ion channels are critical for detection of noxious stimuli in primary sensory neurons. Transient receptor potential (TRP) channels mediate pain sensations and promote a variety of neuronal signals that elicit secondary neural functions (such as calcitonin gene-related peptide [CGRP] secretion), which are important for physiological functions throughout the body. In this review, we focus on the involvement of TRP channels in sensing acute pain, inflammatory pain, headache, migraine, pain due to fungal infections, and osteo-inflammation. Furthermore, action potentials mediated via interactions between TRP channels and the chloride channel, anoctamin 1 (ANO1), can also generate strong pain sensations in primary sensory neurons. Thus, we also discuss mechanisms that enhance neuronal excitation and are dependent on ANO1, and consider modulation of pain sensation from the perspective of both cation and anion dynamics.

## 1. Introduction

The ability to sense elements of the natural environment (including temperature, pH, pressure, light, and noxious compounds) is critical for survival. Detection and response to environmental agents and stimuli are frequently mediated by receptor-type plasma membrane proteins, particularly ion channels that show versatile function in a range of organisms from prokaryotes to eukaryotes. Relative to G-protein coupled receptors (GPCRs), ion channels can directly impact neural excitation by both sensing natural stimuli and converting these signals into electrical changes to affect the polarization state of the plasma membrane. 

In this review, we focus on several transient receptor potential (TRP) channels that are specifically activated by natural compounds and largely localize to primary sensory neurons. There are three types of nerves in primary sensory neurons, including Aβ-, Aδ-, and C-fibers. Aβ-fibers are myelinated afferent nerves that respond to innocuous mechanical stimuli. Aδ-fibers are also myelinated nerves, but alternatively this nervous pathway responds to rapid noxious stimuli. C-fibers are nonmyelinated nerves involved in slow pain [1]. The TRP channel superfamily comprises six subfamilies: TRP vanilloid (TRPV), canonical (TRPC), mucolipin (TRPML), polycystin (TRPP), ankyrin (TRPA), and melastatin (TRPM). Several TRP channels are expressed in small-size dorsal root ganglion (DRG) and trigeminal ganglion (TG) neurons (C fibers and Aδ fibers) [2]. While TRPV1 and TRPA1 are considered to be the major receptors of this superfamily involved in nociception [3]. In particular, TRPV1 and sensitized TRPA1 are activated by heat and cold, respectively, and as such are important for detection of noxious temperature changes. Recently, TRPM3 involvement in heat sensation was also reported in mice [4]. 

The calcium-activated chloride channel, anoctamin 1 (ANO1, also known as TMEM16A) [5,6,7], was recently reported to be directly activated in DRG neurons by extremely rapid temperature changes that reach noxious ranges [8,9]. ANO1 can also be activated immediately downstream of Gq protein-coupled receptors (GqPCRs), including the bradykinin receptor, as evidenced by direct interaction of ANO1 with inositol trisphosphate (IP_3_) receptors on endoplasmic reticulum membranes [10]. Chloride channels typically function in neuronal suppression in the central nervous system, in part because intracellular chloride concentrations are maintained at low levels by the potassium–chloride co-transporter type 2 (KCC2). However, in DRG neurons, KCC2 expression is either absent or very low, whereas expression of the sodium–potassium–chloride co-transporter type 1 (which is an important molecule in the chloride intake pathway) is high [11]. Thus, chloride efflux through ANO1 activation is a key pathway for generation of neuronal excitation in many primary sensory neurons.

Here, we summarize the physiological significance of TRP and ANO1 channels. First, we describe current understanding of representative ion channels, namely TRPA1, TRPV1, and ANO1 (Part 1). Second, we discuss the multiple functions of TRP channels and ANO1 (Part 2). Finally, we propose the significance of those functions in clinical situations, including headache, migraine, and fungus infection (Parts 3 and 4). 

## 2. Basic Understanding of Ion Channels in Primary Sensory Neurons

### 2.1. TRPA1

#### 2.1.1. TRPA1 Activation by Natural Ligands

TRPA1 is activated by many natural ligands such as allyl isothiocyanate (AITC), tetrahydrocannabinol, cinnamaldehyde, allicin, diallyl sulfide, carvacrol, eugenol, gingerol, methyl salicylate, capsiate, thymol, propofol, 1,4-cineole, oleocanthal, and carbon dioxide, and by membrane extension and intracellular alkalization [12,13,14,15,16,17,18,19,20,21,22,23,24,25,26]. Moreover, TRPA1 is activated by calcium [27]. The magnitude of TRPA1 currents gradually increases during application of a given agonist, as observed for the lagging peak current induced after application of -eudesmol from hops to HEK293 cells expressing human TRPA1 [28]. Although the precise mechanism of TRPA1 activation remains unclear, covalent protein modification is involved. Carbons of AITC and *N*-methyl maleimide covalently bind to cysteine in the N-terminus of TRPA1 to enhance channel activation, whereas C-terminal lysine and arginine are important for AITC-mediated activation [29,30,31]. Menthol also has agonistic effects on human and mouse TRPA1, although the effects are bimodal [32,33]. The agonistic and antagonistic effects on mouse TRPA1 involves serine 876 and threonine 877 in the transmembrane (TM)-5 region [33]. Interestingly, G878 is also important for TRPA1-mediated cold sensitivity in rodents [34]. Rodent TRPA1 can be activated by cold stimulation and is involved in cold hyperalgesia after application of complete Freund’s adjuvant (CFA) [35,36]. Although human TRPA1 does not show a cold response, it nonetheless responds to cold at approximately 18 °C if oxidization with dehydroxylation at proline 394 occurs [37].

#### 2.1.2. TRPA1 in Pathological Conditions

TRPA1 activation induces hyperalgesia during inflammation because inflammatory factors (such as bradykinin released by tissue injury) activate and sensitize TRPA1 in DRG neurons. In this pathway, protein kinase A (PKA) and phospholipase C (PLC) are important for TRPA1 sensitization [38]. Adenosine triphosphate (ATP) is another important inducer in inflammatory pain. Pain sensations are enhanced via a similar pathway through activation of purinergic P2Y receptors expressed in DRG neurons. Moreover, P2X receptors are involved in neuropathic pain via phospholipase A2 (PLA2) signaling [39], which activates protein kinase C (PKC), and in turn sensitizes TRPA1 [40].

TRPA1 expression increases after application of nerve growth factor (NGF) and is inhibited by the p38 mitogen-activated protein kinase (MAPK) inhibitor, SB203580. In DRG neurons, NGF released from inflamed tissue phosphorylates p38, which subsequently enhances TRPA1 expression [36]. Thus, the NGF–p38 MAPK–TRPA1 axis is one of the pathways that exacerbates TRPA1-mediated pain sensation in DRG neurons. For example, gastric distension-induced visceral pain relies on activation of both TRPA1 and p38 [41]. TRPA1 localization can be modified by pathological stimulation. TRPA1 localization to the plasma membrane is enhanced in forskolin-treated DRG neurons [42]. Further, in mice, full-length TRPA1 positively translocates to the plasma membrane by co-expression of a TRPA1 splicing variant [43]. There are two TRPA1 splicing variants: TRPA1a is the full-length protein whereas TRPA1b lacks exon 20, which encodes part of TM2 and the intracellular domain between TM2 and TM3. TRPA1b has no ion channel activity but instead enhances TRPA1a translocation to the plasma membrane. One-day of CFA treatment or partial sciatic nerve ligation (PSL) causes inflammatory and neuropathic pain, respectively. In both cases, TRPA1a expression levels increase transiently. Interestingly, TRPA1b expression levels significantly increase while TRPA1a expression reduces to basal levels at five days after CFA treatment or PSL. As such, up-regulation of TRPA1a translocation via TRPA1b overexpression causes a continuous pathological condition.

As with CFA, lipopolysaccharide (LPS) is also often used to induce the inflammatory condition. LPS can activate Toll-like receptor (TLR)-4, and cause subsequent release of multiple cytokines from immune cells [44]. The cytokine, tumor necrosis factor-alpha (TNF-α), enhances AITC-induced calcium increases in nodose and jugular ganglion neurons from rats [45]. However, a recent report suggested that LPS-induced calcium increases in nodose ganglion neurons from mice do not depend on TLR4, even although the responses are reduced in TRPA1 knockout mice [46]. Altogether, these results suggest that LPS directly activates TRPA1. Further, LPS increases single channel activity in TRPA1-expressing CHO cells. Ultimately, this novel relationship between bacteria and primary sensory nerves suggests that TRPA1 antagonists could be valuable for reducing pain induced by bacterial infections.

#### 2.1.3. TRPA1 Activation by Reactive Oxygen Species and Hypoxia

In addition to thermal stimuli and environmental agents, TRPA1 is activated by reactive oxygen species (ROS) such as hydrogen peroxide (H_2_O_2_) [47,48,49]. Responses to certain pathological conditions involving increased ROS synthesis (such as dysesthesia in ischemia and reperfusion of blood flow) are dependent on TRPA1 activity in mice [50]. Pain-related behavior due to dysesthesia is reduced by activation of prolyl hydroxylase (PHD)-2 involving hydroxylation at proline. Under normoxic conditions, TRPA1 steady status activity is maintained by PHD-mediated hydroxylation of proline 394, but under hypoxic conditions hydroxylation is inhibited and H_2_O_2_-induced TRPA1 activity is enhanced [50]. In contrast, high concentrations of oxygen also activate TRPA1 by directly modifying TRPA1 cysteines [51]. Collectively, these two functions allow TRPA1 to act as an oxygen sensor under both hypoxic and hyperoxic conditions.

Side effects of the anti-cancer agent oxaliplatin include induction of various dysesthesias, including peripheral nerve disorder and cold hyperalgesia. These dysesthesias are associated with enhanced TRPA1 expression in DRG and PHD-induced modification of TRPA1 [52,53,54]. Moreover, the oxaliplatin degradation product, oxalate, inhibits PHD and subsequently TRPA1 dehydroxylation, and also promotes cold hypersensitivity upon activation of TRPA1 in response to ROS production by mitochondria [37,55]. Mechanical allodynia associated with oxaliplatin treatment can be inhibited by the TRPA1 antagonist, ADM_09 [56]. Together, these results clearly indicate the importance of the relationship between TRPA1 and the PHD cascade, and also that TRPA1 could be targeted as part of treatment for dysesthesia induced by ischemia and hypoxia, as well as drug-induced cold and mechanical hyperalgesia.

#### 2.1.4. pH Sensing by TRPA1

The relationship between oxygen and pH is physiologically important because hypoxic conditions caused by ischemia induce intracellular acidification. Neuronal death may occur upon low levels of oxygen and glucose, as well as excessive release of glutamic acid from astrocytes, which induces a fatal calcium influx in neurons. Intracellular pH of astrocytes is drastically reduced by lactic acid production due to anaerobic respiration in response to hypoxia. Subsequent acidification induces glutamic acid exocytosis led to brain damage [57]. Importantly, TRPA1 expressed in astrocytes may be activated by acidification [58,59,60]. In addition, activation of TRPA1 expressed in oligodendrocytes can damage myelin [61]. In contrast, intracellular alkalization also affects TRPA1 activity [26]. TRPA1 is activated at approximately pH 8.0, and the alkalization-induced pain-related behavior is significantly reduced in TRPA1 deficient mice. Consequently, the pH dependency of TRPA1 may be beneficial target for the treatment of central nervous system diseases, not only pain. 

#### 2.1.5. Neural Networks Involving TRPA1-Mediated Pain Sensation

A-fiber and C-fiber primary sensory nerves govern fast and slow responses to pain, respectively. Aδ-fibers (mid-sized DRG neurons) innervate lamina I and V, whereas C-fibers (small-sized DRG neurons) innervate lamina I and II of the dorsal horn of the spinal cord [1]. C-fibers also contain peptidergic and nonpeptidergic neurons. Peptidergic neurons contain substance P and CGRP, with both peptides released upon neural excitation. TRPA1-positive neurons are immunoreactive for CGRP in DRG neurons [15]. In healthy mice, these CGRP-positive neurons enhance heat sensation and suppress cold sensation [62]. These findings suggest that TRPA1 in CGRP-positive DRG neurons contributes less significantly to noxious cold sensation.

Neural transmission in the spinal cord can modify pain perception. Substantia gelatinosa (SG) neurons in lamina II are important targets for investigation of how pain sensations are transmitted from the periphery to the central nervous system. Initial understanding on in vivo SG neuronal responses to peripheral stimulation is that SG neuronal activity mediated through non-*N*-methyl-*D*-aspartate (non-NMDA) receptors is enhanced by mechanical stimuli, such as pinch and air flow, but not thermal changes [63]. However, excitatory postsynaptic currents enhanced by capsaicin treatment are detected in approximately 80% of SG neurons in slice patch-clamp recordings [64]. Interestingly, there is no neuronal response to AITC alone (Figure 1). 

Since AITC responses depend on both NMDA and non-NMDA receptors [65], TRPA1-mediated pain signals are likely integrated with TRPV1-mediated pain signals in lamina II of the spinal cord. Importantly, there are three types of DRG neurons: those that express both TRPA1 and TRPV1, TRPA1 alone, or TRPV1 alone. Meanwhile, one study demonstrated that spinal TRPA1 activation by intrathecal administration of the acetaminophen metabolite, N-acetyl-p-benzoquinone imine, enhanced anti-nociception in the spinal cord of mice [66]. Therefore, components of TRPA1-mediated neural systems may participate in pain reduction, while nociception by TRPA1 activation can function in central termini of DRG neurons. For instance, TRPA1 activation by hepoxilin causes mechanical allodynia in rats, whereas pinch-evoked SG neuronal excitation is reduced by increases in inhibitory postsynaptic currents mediated by TRPA1 activation in vivo [67,68]. Thus, consideration of TRPA1 activation in the central nervous system may also be important for investigating pain mechanisms.

#### 2.1.6. TRPA1 Activation by microRNA in the Central Nervous System

Although TRPA1 is expressed in both brain and spinal cord cells, whether it is activated via direct or indirect pathways is unclear. In the central nervous system, only one obvious possibility exists for direct activation of TRPA1 by an endogenous ligand, namely microRNA (miRNA). Among the miRNAs present in cerebrospinal fluid, increased levels of let-7b are particularly associated with the incidence of Alzheimer’s disease. Meanwhile, astrocytes can exacerbate symptoms associated with amyloid-induced TRPA1 activation [69,70]. Let-7b activates TLR7, which is followed by cytotoxicity and direct activation of TRPA1 [71]. Importantly, let-7b release is enhanced by formalin application to DRG, while let-7b injection induces both nociceptive behavior and mechanical allodynia, which are reduced in TRPA1- and TLR7-deficient mice. These results demonstrate a relationship between let-7b and TRPA1 as a possible molecular mechanism of inflammatory symptoms in central nervous system diseases.

### 2.2. TRPV1

Among TRP channels expressed in primary sensory neurons, TRPV1 is well-known for its role in pain [3]. TRPV1 is mainly expressed in small DRG and TG neurons and is activated by capsaicin, capsiate, camphor, allicin, 2-aminoethoxydiphenyl borate, anandamide, N-arachidonoyl dopamine, resiniferatoxin, nitrogen oxide, low pH, noxious heat, hypertonicity, and the double-knot toxin in tarantula venom [14,72,73,74,75,76,77,78,79,80,81]. Expression levels of TRPV1 are enhanced by NGF receptor activation in DRG neurons, while TRPV1 is transported to peripheral termini [82]. TRPV1 activation is enhanced upon phosphorylation by PKA and PKC via A-kinase anchor protein in DRG neurons activated by GPCRs [83,84,85,86]. Due to the lowered thermal threshold of phosphorylated TRPV1 at body temperature, allodynia can be caused by inflammatory pathways depending on GqPCR activation. Many inflammatory factors, including prostaglandin E2, adenosine, ATP, bradykinin, protease, and NGF, are released following tissue injury, with protein kinases activated downstream of each GPCR [1]. In addition, the ionotropic ATP receptor, P2X, may also be involved in modulating activity of protein kinases that target TRP channels, as evidenced by activation of cytosolic PLA2 by P2X3 and P2X2/3 receptors during neuropathic pain [39]. Activated PLA2 can in turn activate PKC to promote phosphorylation of TRP [40]. Taken together, these findings suggest that TRP channel phosphorylation may be caused by both P2X and metabotropic P2Y receptor activation in primary sensory neurons. 

### 2.3. Anoctamin 1

ANO1 is a calcium-activated chloride channel [5,6,7]. Although the ANO family includes ten subtypes, only ANO1 and ANO2 exhibit marked activity as calcium-activated chloride channels, and conductance of ANO1 is larger than ANO2. The crystal structure of fungal ANO has recently been determined at high resolution [87]. Furthermore, the dimer structure of mouse ANO1 (which contains ten TM regions in one subunit) has been clarified by cryo-electron microscopy [88,89]. Accordingly, the calcium binding site was shown to be encompassed by TM6 to TM8. Interestingly, each ANO1 subunit has one pore region surrounded by TM3 to TM8. Structural analysis showed that the dynamic movement of TM6 may be critical for calcium-mediated channel opening.

Although ANO1 can be activated by global increases in intracellular calcium via activation of voltage-gated calcium channels, ANO1 activation through GqPCR is also likely to be important due to direct interactions between ANO1 and IP_3_R in endoplasmic reticulum calcium stores [10,90]. Therefore, ANO1 is possibly involved in nociception induced by inflammatory factors such as bradykinin [91]. ANO1 is also activated by noxious heat in DRG neurons and induces a burning pain sensation [8]. These characteristics may explain why chloride channel activity evokes neural excitation in primary sensory neurons, in that higher intracellular chloride concentrations can be maintained as the equilibrium potential in these cells is more positive than the resting potential in DRG neurons [11]. 

## 3. Collaboration of Ion Channels

Although each ion channel, including TRP channels, independently work as detectors of nociceptive stimuli, some ion channels make physical or functional complexes that are critically involved in pain sensation. In this part, we summarize ion channel interactions and nociceptor populations according to recent reports (Figure 1). 

### 3.1. TRPV1–TMEM100–TRPA1 Interaction

TRPV1 is co-expressed with TRPA1 in DRG neurons. Since TRPA1 activity is enhanced by intracellular calcium, it had been thought that calcium influx through TRPV1 activation could affect TRPA1 function. However, the TRPV1 entity reduces the probability of TRPA1 ion channel opening accelerated by mustard oil [92]. It appears that TRPA1-associated pain is normally reduced by TRPV1 expression, which may be prevented by TMEM100 [92]. TMEM100 is a small membrane protein, and its expression pattern highly overlaps with CGRP. Interestingly, TRPA1 almost co-localizes with TRPV1, TMEM100, and CGRP in DRG neurons (Figure 1). Together, TRPV1, TRPA1, and TMEM100 form a complex, and the interaction between TRPV1 and TRPA1 is suppressed by interposition of TMEM100. Furthermore, a mutant peptide of TMEM100 (T100-Mut) can permeate the plasma membrane and disturb correct binding of TMEM100, thereby inhibiting TRPA1-associated pain-related behavior. This may provide a novel strategy for reducing pain sensation.

### 3.2. TRPV1–ANO1 Interaction

Approximately 80% of TRPV1-positive DRG neurons also express ANO1 by immunostaining [8,93]. To examine the function of this co-expression, we investigated whether these ion channels interact in a setting of acute pain induced by capsaicin. We found that capsaicin-induced currents in isolated DRG neurons were suppressed by the ANO1 inhibitor, T16Ainh-A01 [94]. Although capsaicin-induced action potentials were also inhibited by T16Ainh-A01, inhibition was observed in response to second application of capsaicin (10 min after first capsaicin application). Some DRG neurons may not have exhibited second action potentials, yet there were DRG neurons that did show action potentials in the second application [93]. Another study showed that TRPV1 is desensitized by calmodulin binding. Moreover, TRPV1 function spontaneously and fully recovered after one hour, while desensitization was inhibited by TRPV1 phosphorylation [85]. Accordingly, the second response in some neurons is thought to depend on random phosphorylation levels in each neuron. Nonetheless, T16Ainh-A01 almost completely inhibits these second action potentials. These results suggest that chloride efflux elicited by ANO1 activation may accelerate depolarization to induce secondary action potentials, and that ANO1 inhibition may be effective at reducing pain sensation. In fact, capsaicin-induced pain-related behavior in mice is inhibited by T16Ainh-A01 [93]. Taken together, these findings indicate that the TRPV1–ANO1 interaction is critical for sensation of noxious stimuli.

ANO1 is also co-expressed with TRPV1 in TG neurons and is functionally involved in heat sensation [95,96]. In addition, TRPV1 and ANO1 expression levels are enhanced by estrogen in female rats [97]. Based on these observations, the TRPV1–ANO1 interaction may be a crucial target for pain therapies. According to a previous report, approximately 70% of ANO1-positive neurons do not colocalize with CGRP [8]. Although contribution of the TRPV1–ANO1 interaction in CGRP release is unknown, the TRPV1–ANO1 interaction may encompass the alternative side of the TRPV1–TMEM100–TRPA1 interaction system (Figure 1).

### 3.3. Triple Conjugation of TRPV1, TRPA1, and TRPM3 

TRPM3 is a heat sensitive TRP channel that functionally couples with TRPV1 and TRPA1 [98]. Although TRPM3-deficient mice show a delayed tail flick at 57 °C, the effect of TRPM3 alone on heat sensation is unclear because tail flick behavior induced at 57 °C in TRPM3/TRPA1 double-deficient mice is no different to wild-type mice [4]. However, triple conjugation of TRPV1, TRPA1, and TRPM3 is important for detecting the noxious heat environment [4]. Withdrawal latency of TRPM3-deficient mice in the hot-plate test (50 °C) is the same as in wild-type mice [98]. Interestingly, this behavior disappears in TRPV1/TRPA1/TRPM3 triple-deficient mice, while the other responses to nociceptive stimuli are normal. Furthermore, wild-type and triple-deficient mice show a similar distribution on a gradient temperature plate (from 5 to 50 °C). In addition, CGRP-expressing DRG neurons are involved in heat sensation [62], and CGRP release from skin preparations is enhanced by the TRPM3 agonist, CIM0216, which is the same as for capsaicin treatment [99]. These findings indicate that the likely multiple function of TRPV1, TRPA1, and TRPM3 in peptidergic DRG neurons is to escape from a noxious heat environment.

## 4. Headache and Migraine

Primary headaches (i.e., those that are not associated with another disorder) are one of the most common causes of disability worldwide. The various types of headache include migraine, tension, and trigeminal autonomic cephalalgia [100]. Migraine is a multifactorial and incapacitating neurovascular disorder characterized by recurrent attacks of severe, unilateral, and throbbing headache, which can be aggravated by routine physical activity and can last from several hours to several days [100]. Migraine attacks often involve not only head pain, but also several premonitory and postdromal symptoms that occur before the headache initiates and persist after the headache ends, respectively. These symptoms are diverse and may include hypersensitivity to light, sound, smell, fatigue, neck stiffness, yawning, mood change, nausea, vomiting, cutaneous allodynia, and transient visual disturbances termed aura [101,102,103,104]. Migraine attacks can be triggered by many internal and external stimuli such as stress, hormonal fluctuations, sleep disturbances, skipping meals, weather changes, and ingestion of alcoholic beverage or certain types of food [105,106]. This multifactorial origin, as well as the variety of symptoms, have complicated identification of the underlying mechanisms of migraines. Although the events that trigger migraines remain unknown, the pain phase of migraine headaches is thought to involve activation and sensitization of primary afferent nociceptors that innervate the dural and meningeal vasculature. Indeed, the trigeminovascular system (TGVS) is a key component in pain initiation and transmission in migraine. Specifically, perivascular TG nerve endings are known to release CGRP, which induces vasodilation of cranial blood vessels and degranulation of meningeal mast cells, leading to neurogenic inflammation [107,108]. Interestingly, recent evidence indicated that a variety of ion channels, including TRP channels, make important contributions to migraine physiopathology (Figure 2 and Table 1). Several recent reviews discussed involvement of TRP channels in migraine, confirming significant interest in these channels as molecular targets for treatment of migraine [109,110,111]. In the following sections, we will discuss recent findings regarding TRP channels in migraine. 

### 4.1. TRPV1 as a Crucial Migraine Initiator

The first hint that TRPV1 is involved in migraine was demonstration of its co-expression with CGRP in rat TG neurons [134,135] and mouse dural afferent neurons [136], suggesting a crucial role for TRPV1 in migraine. Moreover, in rat dura mater, application of capsaicin is accompanied by vasodilatation mediated by CGRP release from sensory afferent nerves [112]. Another substantial link between migraine and TRPV1 was the demonstration that a frequent migraine trigger, ethanol, induced neurogenic vasodilation via TRPV1 activation and subsequent CGRP release in the TGVS of guinea pigs [118]. Although the triggering event that actually initiates a migraine attack remains elusive, Meents et al., proposed that the premonitory aura exhibited by some migraineurs promotes endogenous activation of TRPV1 [137]. Aura arises from a phenomenon called cortical spreading depression (CSD), a short-lasting depolarization of cortical neurons that is known to increase extracellular concentrations of H^+^, which can ultimately activate TRPV1 and induce CGRP release [119]. However, not all migraineurs experience aura, indicating that other endogenous mechanisms likely contribute to TRPV1 activation within the dura. Hence, although TRPV1 is currently recognized as a key player in migraine initiation, it is also likely involved in other phases and characteristics of migraine. 

Sensitization of peripheral and central trigeminovascular neurons is usually observed following migraine attack onset. Peripheral sensitization mediates the throbbing perception of a headache, whereas sensitization of second-order neurons from the spinal trigeminal nucleus mediates cephalic allodynia and muscle tenderness [138]. Sensitization of TG neurons may contribute to direct sensitization of TRPV1, either by its increased activity or translocation to the cell membrane, or by increased protein production. Interestingly, the cerebrospinal fluid of chronic migraineurs (i.e., patients who have more than 15 migraine attacks per month) exhibit elevated levels of inflammatory mediators, including NGF [139]. Bradykinin and prostaglandin E2 are inflammatory mediators also released during neurogenic inflammation, and are commonly used in an animal model of headache to induce a chronic state of trigeminal hypersensitivity [113,114,115]. As already noted, NGF can trigger TRPV1 translocation to the plasma membrane, while bradykinin and prostaglandin E2 can orchestrate TRPV1 phosphorylation, which lowers its activation threshold. Moreover, TRPV1 expression is up-regulated in nerve fibers that innervate arteries in the scalp of chronic migraine patients [116]. Therefore, neurogenic inflammation that occurs during migraine attacks likely contributes to sensitization by modulating TRPV1 channel activity and expression.

Schwedt and colleagues proposed an interesting idea, namely that a state of persistent sensitization is maintained in migraineurs that enable more ready firing of TGVS [117]. They hypothesized that a cyclical process of migraine headaches causes interictal sensitization that contributes to predisposition to future migraine attacks. In their study, episodic and chronic migraineurs displayed enhanced sensitivity to thermal stimulation (decreased heat and cold pain threshold and tolerance) during the interictal period, compared with non-migraine controls. Such sensitization can partly be attributed to TRPV1 and may also explain why migraine patients do not tolerate ambient temperature changes [140]. In the same manner, a recent study showed that migraine patients exhibit enhanced extracephalic capsaicin-induced pain sensation during interictal periods, supporting the contribution of TRPV1 to interictal sensitization [141].

Another characteristic of migraines is their difference in prevalence and perception between men and women. Women are three-times more likely to suffer from migraine than men, and women experience more frequent, longer-lasting, and more intense migraine attacks than men [142]. Higher estrogen levels may in part be responsible for these differences. Interestingly, estrogen was recently shown to increase pain sensation by up-regulating expression of TRPV1 and ANO1 in TG neurons from female rats [97]. This could explain why women exhibit a higher susceptibility to migraine, and suggests that potential interaction of TRPV1 and ANO1 in TG neurons may be involved in migraine initiation.

Obesity has also been linked to migraine prevalence since it increases the risk of developing a migraine. Further persons with obesity suffer from more frequent and severe headache attacks [143]. A study involving mice fed a high-fat diet showed that facial intradermal injection of lower capsaicin doses are needed to induce photophobic behavior in obese mice compared to non-obese control mice [120]. Also, cell size distribution among TRPV1-positive cultured TG neurons from obese mice shifted towards larger cell diameters compared with control mice, and a higher capsaicin-induced calcium influx was observed in these neurons. In another mouse model that induced obesity through feeding of a high-fat and high-sucrose diet, dural application of capsaicin induced enhanced vasodilatory and vasoconstrictor responses compared with control animals. Basal and capsaicin-induced CGRP release from meningeal afferents was also increased [144]. These findings may explain why diet-induced obesity is associated with TGVS sensitization, which might occur via TRPV1 modulation, although the precise molecular mechanism is unclear.

### 4.2. TRPA1 as an Intriguing Migraine Contributor

TRPA1 is also co-expressed in CGRP-positive nociceptors [136,145] and has attracted significant attention in the context of migraine pathophysiology due to its sensitivity to numerous exogenous and endogenous compounds. Indeed, environmental irritants such as cigarette smoke or formaldehyde, as well as ROS and reactive nitrogen species (RNS), can activate TRPA1 (for review see [123]). A link with migraine can subsequently be readily deduced from the ability of these compounds to generate headache. Appropriately, TRPA1 activation in trigeminal nerve endings located in the nasal mucosa are suspected to trigger headache when irritants are inhaled. Indeed, intranasal administration of irritant compounds in rats can induce CGRP release via TRPA1 activation and increase cerebral blood flow [122]. Similarly, intranasal administration of umbellulone, the volatile active compound from *Umbellaria californica*, known as the “headache tree”, evokes TRPA1-mediated and CGRP-dependent neurogenic meningeal vasodilation in mice [146,147]. In contrast, compounds known for their anti-headache properties were shown to desensitize TRPA1. For example, stimulation of rat TG neurons with parthenolide, a compound extracted from the feverfew herb (*Tanacetum parthenium* L.), which has been used for centuries to reduce pain, fever, and headaches [148]), induced potent and prolonged desensitization of TRPA1 channels, which rendered peptidergic neurons unresponsive to any stimulus and unable to release CGRP [149]. Similar properties were observed for isopetasin, a major constituent of extracts from butterbur, a plant known to have anti-migraine effects. Isopetasin visibly desensitized TRPA1 in patch-clamp experiments with rat TG neurons, while it also inhibited nociception and neurogenic dural vasodilatation mediated by TRPA1 in vivo [150]. 

Another important migraine trigger is ROS. Several studies reported increased oxidative stress in migraine patients both during headache attacks and in the interictal period (the period between migraine attacks) [151,152,153]. As already noted, ROS are potent TRPA1 activators, and in a recent study were shown to mediate the CSD responsible for aura [154]. In that study, exogenous H_2_O_2_ activated TRPA1 expressed in cortical neurons in mice brain slices, raising their susceptibility to CSD. Conversely, endogenous ROS produced upon CSD development [155] activated TRPA1 expression in TG neurons and mediated CGRP production, leading to a positive feedback loop that regulates cortical susceptibility to CSD. Based on these findings, it was proposed that reducing ROS production together with blockade of neuronal TRPA1 could help prevent stress-triggered migraine.

RNS can also act as TRPA1 agonists [79], and have been linked to headaches and migraine development. Indeed, an increase in endogenous nitric oxide (NO) production is observed during migraine attacks [156]. Eberhardt and colleagues reported that nitroxyl, generated by a redox reaction between NO and hydrogen sulfide can trigger TRPA1 activation in the TGVS, leading to CGRP release in the cranial dura mater of rats [145]. This pathway ultimately resulted in vasodilation and increased meningeal blood flow, and could also account for the headache phase of a migraine attack. Similarly, the well-known headache inducer, glyceryl trinitrate, targets TRPA1 in TG neurons to generate periorbital oxidative stress and mechanical allodynia [157].

### 4.3. TRPM8 as a Familial Migraine Instigator

TRPM8 is found on both Aδ and C fiber afferents, and is important for the activation of peripheral sensory neurons by cold temperature. It is activated at non-noxious cold temperatures (< 26 °C) and by compounds that produce a cooling sensation such as menthol or eucalyptol [158,159]. While its role as a cold sensor has been firmly established, it is not the case regarding its role in pain sensation. It is still under debate whether TRPM8 reduces or exacerbates pain sensation, and the most recent view on the matter is that TRPM8-expressing afferent fibers have the ability to both produce and alleviate pain, and the outcome will be determined by context (see for review [133,160]). As such, TRPM8 has begun to gather attention in the migraine field. A genetic predisposition to migraine is well-recognized: migraineurs presenting a hereditary component account for 42% of patients with migraine, as shown in studies on families and twins [161,162]. Migraine is genetically complex because many genetic variants with small effects and environmental factors can confer migraine susceptibility [163]. However, several genome-wide association studies from different cohorts identified single nucleotide polymorphisms (SNPs) in the gene encoding TRPM8, suggesting an important role for this TRP channel in migraine pathophysiology [128,129,130,131,132]. Several of these variants are located in regions involved in transcriptional regulation and may therefore impact upon TRPM8 expression levels. Moreover, in calcium imaging experiments, some TRPM8 SNP variants heterologously expressed in HEK293 cells showed alterations of channel functionality [164]. Based on these results, TRPM8 variants identified in migraine patients likely contribute to migraine pathology. In adult mice, TRPM8 is also expressed in dural trigeminal nerve endings, albeit rather sparsely [136,165]. Age-dependent decreases in TRPM8 expression in TG neurons appears to play a role in pathways that are differentially regulated with age, in that both the density and number of branches of TRPM8-expressing fibers are comparable to CGRP-expressing fibers in postnatal mouse dura. Specifically, both are reduced by half in adult mouse dura [165]. However, the functional consequence of this reduction remains unclear.

Although TRPM8 is a well-established cold transducer, limited temperature fluctuations in the skull suggest that this activity is less important in dural tissue. Thus, endogenous TRPM8 activators within the dura are unknown. Similarly, whether TRPM8 activation within the dura has a pro- or anti-nociception effect is unclear. The most recent studies yielded opposite results. Ren and colleagues observed that dural application of menthol resulted in inhibition of nocifensive behavior in a mouse migraine model induced by inflammatory mediators, suggesting an anti-nociceptive effect of TRPM8 [165]. In contrast, dural application of icilin produced migraine-like behavior in mice, such as cutaneous facial and hind paw allodynia. Pretreatment with the TRPM8 antagonist, AMG1161, attenuated these behaviors [133]. The contrary results obtained in these two studies may be due to the model used: when activated alone, dural TRPM8 appears to have a pro-nociceptive effect, but when activated together with inflammatory mediators, TRPM8 has an anti-nociceptive effect. Ultimately, TRPM8 activation may act as a migraine initiator in the first instance, but have another role during the neurogenic inflammation phase. Moreover, as suggested by Dussor and Cao, these different outcomes might also reflect how TRPM8-expressing fibers project to central neurons as well as the context dependence of TRPM8 activation [166]. More studies are needed to fully understand the role of TRPM8 in dural afferents and migraine pathophysiology.

### 4.4. TRPV4 as an Indirect Migraine Modulator

TRP Vanilloid 4 (TRPV4) is a widely distributed cationic channel that participates in the transduction of both physical (osmotic, mechanical, and heat) and chemical (endogenous, plant-derived, and synthetic ligands) stimuli (see for review [126]). As a mechanosensitive channel, TRPV4 has attracted increasing interest in the context of migraine. Indeed, headaches can be influenced by changes in intracranial pressure. Recently, TRPV4 was shown to be expressed in dural afferents, and its activation in the dura of freely-moving rats could produce migraine-like behavior such as cutaneous allodynia [125]. Although dural afferents are known to be mechanically sensitive, whether TRPV4 activation that contributes to migraine is due to mechanical stimulation or another endogenous mechanism remains to be elucidated. Another study showed that both TRPV4 and TRPA1 can be activated by the irritant formalin in the TGVS, and result in downstream MEK–ERK pathway activation and pain behavior in mice [127]. However, whether formalin directly or indirectly activates TRPV4 is unknown. Nevertheless, taken together, these findings suggest that TRPV4 could also be a promising target for agents that provide relief from pain that originates in the trigeminal system.

### 4.5. TRP Channel Modulators for Acute Treatment of Migraine Attacks

As an initial line of investigation, TRPV1 agonists were considered as potential analgesics to treat headaches. Intranasal applications of civamide and capsaicin were reported to alleviate headache pain during migraine attack [167,168]. However, in most patients these agents caused severe side effects, such as nasal burning and lacrimation, and thus impeded their clinical use for the treatment of acute migraine. Instead, TRP channels antagonists show more promise as a novel approach to prevent or treat acute migraine attacks. However, an initial clinical randomized trial conducted in 2009 showed that TRPV1 channel blockers failed to treat migraine attacks. In this study, the compound SB-705498 did not relieve headache pain for up to 24 h post-dose [169]. Although this outcome does not exclude a contribution of TRPV1 to migraine pathology, it indicates that selectively targeting TRPV1 alone is not sufficient for acute treatment of migraine attacks. Another explanation for the lack of success with TRPV1 channel blockers is that SB-705498 is largely ineffective in humans. Several other clinical trials using this compound showed no or poor efficacy for treating different conditions. Notably, SB-705498 did not relieve itching arising from histamine-induced pruritus, prevent coughing in refractory chronic cough, or alleviate symptoms elicited by cold, dry air in non-allergic rhinitis [170,171,172], despite documented involvement of TRPV1 in these disorders. To date, no other clinical trials using TRP channel antagonists have been performed for migraine, but numerous in vivo studies show their potential for the development of new therapeutic strategies. Indeed, in a recent study, two TRPV1 antagonists, JNJ-38893777 and JNJ-17203212, reduced or even completely abolished capsaicin-induced CGRP release from TG neurons in two different animal models of migraine [121]. These compounds, used alone or together with other blockers of important molecular players, could be promising pain relief medicines. 

Interestingly, a new molecule, Compound 16-8, which specifically co-targets TRPV4 and TRPA1, was developed based on the TRPV4 antagonist, GSK205 [124]. Compound 16-8 was reported to inhibit both channels at sub-micromolar potency and also abolish formalin-induced trigeminal pain in an in vivo model. This suggests that dual inhibitors may be more effective in treating pain elicited by several molecular players, such as pain that occurs in headaches and migraine induced by irritant compounds. To date, no clinical study has focused on TRPV4. This may be due to limited research concerning TRPV4, and the fact that dual inhibition strategies have not yet been considered for the treatment of migraine. Moreover, although the contribution of ANO1 to pain mechanisms in TG neurons is not fully elucidated [96] and interactions between TRP channels and ANO1 await investigation in the TGVS, we contend that ANO1 in TG neurons likely behaves similarly to that seen in DRG neurons. Thus, simultaneous blockage of TRP and ANO1 channels has potential to provide strong pain relief from headache.

Although TRPM8 variants are associated with migraine susceptibility, whether therapeutic strategies that target this channel should be agonists or antagonists, is unclear. As such, additional information about the role of TRPM8 in migraine development is needed before new therapeutics that focus on this channel can be pursued. 

## 5. Infection and Immunity

Pain sensation is a negative stress for animals, and CGRP release from nociceptors exacerbates symptoms. Conversely, we recently clarified that CGRP release dependent on TRPV1 and TRPA1 activity in DRG neurons is involved in bone protection during fungus infection. This phenomenon is supported by several physiological mechanisms, including ATP release from keratinocytes, neural excitation of sodium channel 1.8 (Nav1.8)-positive DRG neurons, and CGRP-dependent suppression of osteoclasts activated via TNF-. Thus, in this part, we comprehensively describe the physiological and pathological systems involved in cutaneous infection to bone inflammation (Figure 3).

### 5.1. Nociception by C. albicans

*Candida albicans* infections can cause skin or vulvar pain. Breast candidiasis is characterized by severe pain around the nipple [173]. *C. albicans* in the vagina causes itching and mechanical allodynia [174]. *C. albicans* can also enter skeletal tissue and induce painful bone infection [175]. Although *C. albicans* has algesic activity, the mechanisms by which this fungus triggers pain remains completely unknown.

TLR4 expressed on myeloid cells are involved in recognition of fungal mannan and cytokine production upon MyD88 and TIR-domain-containing adapter-inducing interferon-beta (TRIF) activation. The fungal cell wall contains β-glucan and mannan on the intracellular and extracellular face, respectively [176]. Surface exposure of β-glucan is sensed by dectin-1 [177]. Activated dectin-1 assembles as a multimeric complex and induces signaling via an ITAM-like motif, promoting formation of CARD-9–Bcl-10–Malt-1 trimers (CBM trimer) and activation of the NLRP3–ASC–ICE complex (NLRP3 inflammasome). CBM trimers and NLRP3 inflammasome activation are both required to induce cytokine production [178]. 

Candidalysin was recently discovered and is the first fungal cytolytic peptide [179]. This peptide may also contribute to the pathogenesis of fungal invasion. Pain induced by fungal infections is thought to be caused by inflammation, but a recent study suggested that both *Staphylococcus aureus*-derived N-formulated peptides and α-hemolysin can directly stimulate nociceptors [180]. Therefore, nociceptors may be able to sense pathogens, but the underlying molecular mechanisms behind fungal nociception remain unclear. 

In the colonization phase, *C. albicans* forms yeast-like structures that are harmless because colony growth is suppressed by host immunity and the natural antagonistic effects of microbial flora. When the yeast form of *C. albicans* attaches to the skin of an immunocompromised host, budding growth is immediately induced and the soluble β-glucan form is secreted. Notably, β-glucan-induced allodynia is relatively severe compared with that induced by mannan and other pathological components such as Candidalysin. Furthermore, dectin-1-deficient mice are completely unresponsive to *C. albicans* or β-glucan-induced pain in a MyD88/TRIF/inflammasome-independent manner. Moreover, we discovered that *C. albicans* induces acute pain by stimulating Nav1.8-positive nociceptors in primary sensory neurons via the dectin-1-mediated PLCγ2–TRPV1/TRPA1 axis. β-Glucan also induces allodynia, which is dependent on dectin-1-mediated ATP secretion from keratinocytes. Notably, keratinocyte-derived extracellular ATP stimulates sensory neurons via P2X receptors. We also found that mice deficient in the ATP transporter, vesicular nucleotide transporter (VNUT), are unresponsive to β-glucan-induced allodynia, while the VNUT inhibitor clodronate has potent prophylactic potential to target fungal nociceptive symptoms. Together, these findings suggest that ATP- or VNUT-targeted therapies such as clodronate treatment may be a promising therapeutic option for treating pain or allodynia associated with fungal infections [181].

### 5.2. Secondary Symptoms Following Nociception

Nociceptor innervation is seen in skin and bone. Although the function of nociceptors in the osteo–immune system is unclear, ion channels in the DRG may be responsible for sensing noxious stimuli [182]. During inflammation, pro-algesic cytokines derived from immune cells gradually evoke allodynia, leading to production of neuropeptides such as CGRP [183], which in turn causes vasodilatation, impaired insulin release, and enhanced Th17 cell function [184,185,186]. Meanwhile, depletion of TRPV1-positive neurons or CGRP deficiency can lead to osteoporosis [187,188]. Thus, nociceptors may modulate osteo–immune system activity, but how they influence pathogen-induced inflammation and bone destruction in a physiological context remains unknown.

To investigate these questions, we injected LPS or β-glucan into the hind paw of *Nav1.8CreRosa26DTA* mice, a nociceptor-deficient line. Notably, LPS-induced osteo-inflammation was unaffected, suggesting that nociceptors do not affect TLR-induced osteo-inflammation. In contrast, *Nav1.8CreRosa26DTA* mice injected with β-glucan exhibit severe skin inflammation and bone destruction, indicating that nociceptors are negative regulators of fungal osteo-inflammation. Similar to *Nav1.8CreRosa26DTA* mice, TRPV1/TRPA1 double-deficient mice exhibit severe osteo-inflammation in response to β-glucan, and this phenotype was rescued by CGRP administration. Notably, β-glucan injection into the hind paws of both *Nav1.8CreRosa26DTA* and TRPV1/TRPA1 double-deficient mice abolished serum CGRP, indicating that TRP channels acting as nociceptors are required for CGRP induction. To address how CGRP inhibits osteo-inflammation, we assessed the effects of CGRP on osteoclast formation and cytokine production. Intriguingly, we discovered that nerve-derived CGRP inhibits osteoclast actin polymerization via cAMP induction, leading to impaired osteoclast multinucleation. We also found that the CGRP-induced transcriptional repressor, Jun dimerization protein 2, selectively blocks dectin-1-mediated pro-inflammatory cytokine production in myeloid cells via direct inhibition of p65. These unexpected roles for β-glucan-stimulated nociceptors suggests the existence of novel sensocrine pathways that may play a role in fungal osteo-inflammation [181]. 

## 6. Conclusions

Studies from the last two decades show the importance of TRP channels in pain sensations caused by noxious temperatures and many chemicals. In particular, TRPA1 is activated in many conditions and its activity evokes an extremely uncomfortable sensation. Therefore, TRPA1 may be a crucial target for pain treatment, although TRPV1 contribution to more specific nociception cannot be disregarded. While some pharmaceutical companies are already focused on the development of TRPA1 and TRPV1 antagonists [189], ANO1 inhibition may also be effective in treating pain because the role of ANO1 is akin to an amplifier, and its suppression does not generate a painless condition. Namely, a level of nociception that is sufficient enough to sense damage for survival can be maintained in an ANO1-blocked state, but not with shutdown of the detectors, i.e., TRP channels.

Sensory systems do not only detect noxious stimuli but also participate in the establishment of inflammation or chronic pain. Neural excitation induces CGRP release from nociceptor termini, inducing inflammation that can ultimately lead to sensitization of nociceptors. Therefore, inhibition of CGRP release by suppression of TRP channel activity is expected to provide relief for intractable pain, headache, and migraine. However, this strategy may result in dangerous secondary effects in some diseases, including fungus infection. Bone is often disrupted in *Candida* infection, a situation induced by osteoclast activity. It remains unclear why osteoclast activity is up-regulated during infection, but in this case, CGRP release from nociceptors becomes beneficial by suppressing two pathways: TNF-α release from myeloid cells and overactivation of osteoclasts. Thus, inflammation induced by CGRP is not detrimental in some pathological conditions, yet targeting it for pain relief might not always be the best strategy.

In fact, pain sensation negatively controls our physiological conditions, including our emotions. Recent reports indicate that complete abolishment of pain induces a severe pathological condition to our body. Therefore, we believe that the important point of pain management is to decrease pain to a tolerable level, but one that is sufficient enough to maintain natural protection against tissue damage. To develop this strategy, there is a need for the discovery of new TRP channels and ANO1 inhibitors that could be used concomitantly and adjusted depending on each patient’s condition. 

## Figures and Tables

**Figure 1 ijms-20-03411-f001:**
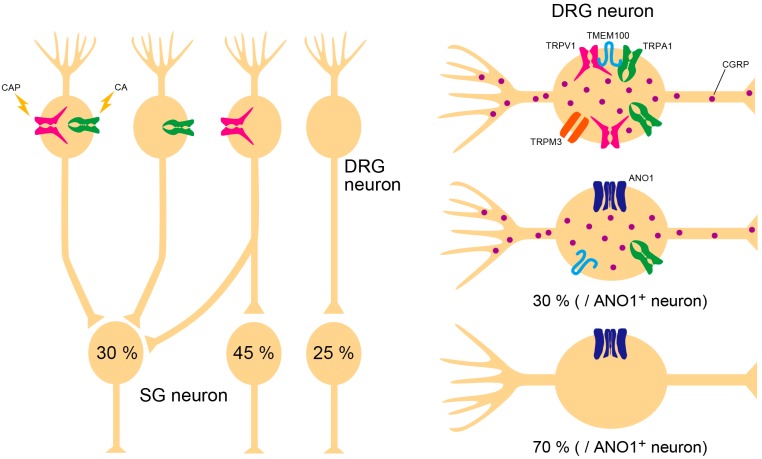
Nociceptor populations in substantia gelatinosa (SG) neurons of lamina II, which receive nociceptive inputs from dorsal root ganglion (DRG) neurons, no neurons respond only to TRPA1-associated stimuli. Approximately 30% of SG neurons are double-positive to capsaicin (CAP) and cinnamaldehyde (CA), 45% of SG neurons response to only CAP, and 25% of neurons show no effect to either CAP or CA. There are calcitonin gene-related peptide (CGRP)-positive and -negative neurons in peripheral sensory nerves. Most TRPV1–TMEM100–TRPA1 complexes and TRPV1–TRPA1–TRPM3 trios are expressed in CGRP-positive neurons. Anoctamin 1 (ANO1) is also expressed in CGRP-positive neurons, however approximately 70% of ANO1-expressing neurons are CGRP-negative.

**Figure 2 ijms-20-03411-f002:**
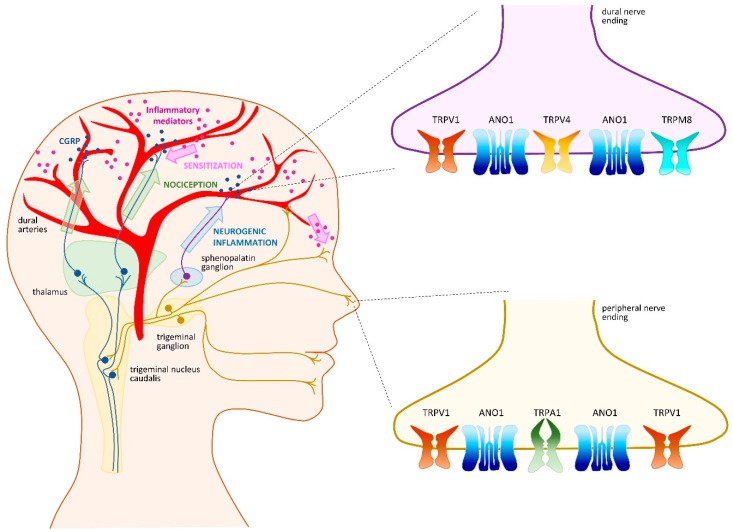
Transient receptor potential (TRP) channels in the trigeminovascular system and migraine development. Migraine triggering events remain unknown and may be a central or peripheral process. The precise order of events contributing to migraine pain is also debated. It is generally admitted that activation and sensitization of primary afferent nociceptors that innervate the dural and meningeal vasculature trigger both calcitonin gene-related peptide (CGRP)-induced vasodilatation and neurogenic inflammation. Pain signals pass through the trigeminal nucleus caudalis, which relays signals to higher order neurons in the thalamus and cortex (green arrows). A trigemino–parasympathetic or trigeminal autonomic reflex arc passes through the sphenopalatine ganglion and is responsible for migraine pain by mediating neurogenic inflammation (blue arrow). Central and peripheral sensitization (pink arrows) may contribute to maintenance of pain signals and predispose to future migraine attacks. Transient receptor potential (TRP) vanilloid 1 (TRPV1) and anoctamin 1 (ANO1) might be involved in initiation, nociception, and sensitization processes of migraine. TRP ankyrin 1 (TRPA1) might be more relevant in the initiation phase. There are few studies available on TRPV4 and TRP melastatin 8 (TRPM8), yet these receptors might be important in pain signal transmission or neurogenic inflammation.

**Figure 3 ijms-20-03411-f003:**
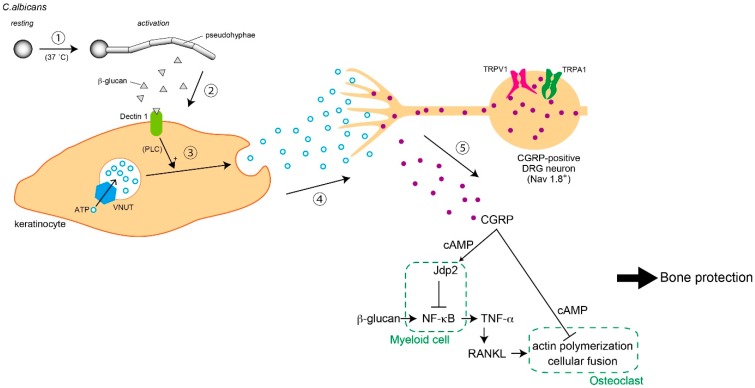
Bone protection system from *Candida* infection. Bone disruption followed by inflammation is worsened by overactive osteoclasts. Nociceptors suppress osteoclast development through calcitonin gene-related peptide (CGRP) release. The process involves: (1) *Candida albicans* is activated in an optimal environment (e.g., body temperature); (2) β-glucan released from pseudohyphae bind to its receptor, dectin-1, on the plasma membrane of keratinocytes; (3) ATP release from keratinocytes is enhanced through the phospholipase C (PLC) pathway; (4) neuronal excitation in voltage-gated sodium channel 1.8 (Nav1.8)-positive dorsal root ganglion (DRG) neurons; and (5) CGRP release from DRG neurons. Jun dimerization protein 2 (Jdp2) is activated by CGRP through a cAMP cascade in myeloid cells. In turn, tumor necrosis factor-alpha (TNF-α release (which accelerates osteoclast development) is suppressed. TNF-α-dependent inflammation is induced by the direct effect of β-glucan on myeloid cells. Furthermore, the CGRP–cAMP axis in osteoclasts also inhibits over-development. Thus, these pathways from skin to bone induce bone protection and inhibit bone inflammation during fungus infection.

**Table 1 ijms-20-03411-t001:** Roles of transient receptor potential (TRP) channels in migraine and their potential as therapeutic targets. RNS: reactive nitrogen species, ROS: reactive oxygen species, TRPA1: transient receptor potential ankyrin 1, TRPM8: transient receptor potential melastatin 8, TRPV1: transient receptor potential vanilloid 1, TRPV4: transient receptor potential vanilloid 4.

	Action	Potential Endogenous Modulators	Antagonists with Potential Clinical Use
TRPV1	CGRP release and vasodilatation [112] Sensitization of peripheral and central neurons (during and between migraine attacks) [113,114,115,116,117]	Alcohol [118] Cortical spreading depression (lowered pH in cortical neurons) [119] Inflammatory mediators [113,114,115] Estrogens [97] Obesity-related mechanisms [120]	JNJ-38893777 [121] JNJ-17203212 [121]
TRPA1	CGRP release and vasodilatation [122]	Volatile irritants [123] ROS and RNS [123]	Compound 16-8 [124]
TRPV4	Cutaneous allodynia [125] Nociception [126,127]	Mechanical force [126] Formalin (directly or undirectly) [127]	Compound 16-8 [124]
TRPM8	Genetic predisposition [128,129,130,131,132] Anti- and pronociception (depending on context) [133]	Unknown	Unknown

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
