# Peer review of "Emerging Perspectives on Pain Management by Modulation of TRP Channels and ANO1"

_ijms, 2019, doi:10.3390/ijms20143411_

Round 1
Reviewer 1 Report
This is a well-written and very comprehensive review to summarize the roles of different TRP channels in thermal pain, migraine, and pain associated with infection. There are only 2 minor points for authors’ consideration.
1. It would be helpful to further describe how intracellular alkalization affects TRPA1 activity and associated pain sensation.
2. The content of the Headaches and Migraine section is excellently organized. For an outstanding review article, it would be good to have a table to summarize the differential roles of TRP channel subtypes in migraine.
Author Response
This is a well-written and very comprehensive review to summarize the roles of different TRP channels in thermal pain, migraine, and pain associated with infection. There are only 2 minor points for authors’ consideration.
We thank the reviewer for their positive comments.
We have modified our manuscript based on the reviewers’ comments.
1. It would be helpful to further describe how intracellular alkalization affects TRPA1 activity and associated pain sensation.
We have added a sentence describing the alkali effect on TRPA1 (p.7, lines 186–189) as below.
" TRPA1 is activated at approximately pH 8.0, and the alkalization-induced pain-related behavior is significantly reduced in TRPA1 deficient mice. Consequently, the pH dependency of TRPA1 may be beneficial target for the treatment of central nervous system diseases, not only pain."
2. The content of the Headaches and Migraine section is excellently organized. For an outstanding review article, it would be good to have a table to summarize the differential roles of TRP channel subtypes in migraine.
We have added a new table and figure 2 in the ‘Headache and migraine’ section.
Reviewer 2 Report
It was difficult for me to understand what the purpose of this review was. After reading the title and abstract when deciding if I would accept the invitation to review, I assumed the manuscript would be a description of interactions between TRP channels and anoctamin 1 (TMEM16A) and how these interactions played a role in pain. However, after reading the article, there is very little about anoctamin 1 aside from a few paragraphs. The second section, a large portion of the manuscript, discusses migraines, and the role TRP channels might play, but aside from one sentence, does not mention anoctamin 1. The final section on infection/inflammatory pain has no mention of anoctamin 1 but does discuss VNUT (the vesicular nucleotide transporter), which is neither a TRP channel or anoctamin 1 related, as a potential target for pain. Therefore, in my opinion, the manuscript suffers from a definite lack of focus and could be greatly improved by narrowing its scope (and potentially changing its title).
Other concerns:
1. The text calls out a Figure 1, but the copy I received had no figure included. On a related note, the manuscript could use many more figures than just one, graphical representations of the pathways discussed would go a long way in helping the reader sift through the massive amount of information the authors discuss in this review. Even a figure depicting the different types of TRP channels and their endogenous and exogenous agonist/antagonists would be helpful for the reader.
2. While the authors discuss TRPV1 and TRPA1 individually in detail, they give very little information about the well-known cross-sensitization that occurs between the two channels.
3. The section on migraines includes a paragraph about TRPM8, which appears relatively out of the blue, i.e. there was no discussion of TRPM8 in the beginning of the paper where TRP channels and their functions are discussed.
4. There are a few places where the authors could improve their use of English grammar.
Author Response
Comments and Suggestions for Authors
It was difficult for me to understand what the purpose of this review was. After reading the title and abstract when deciding if I would accept the invitation to review, I assumed the manuscript would be a description of interactions between TRP channels and anoctamin 1 (TMEM16A) and how these interactions played a role in pain. However, after reading the article, there is very little about anoctamin 1 aside from a few paragraphs. The second section, a large portion of the manuscript, discusses migraines, and the role TRP channels might play, but aside from one sentence, does not mention anoctamin 1. The final section on infection/inflammatory pain has no mention of anoctamin 1 but does discuss VNUT (the vesicular nucleotide transporter), which is neither a TRP channel or anoctamin 1 related, as a potential target for pain. Therefore, in my opinion, the manuscript suffers from a definite lack of focus and could be greatly improved by narrowing its scope (and potentially changing its title).
According to your suggestion, we have modified our explanation regarding the focus of the manuscript in the last paragraph of the Introduction (p.4, lines 82–86) as below.
" Here, we summarize the physiological significance of TRP and ANO1 channels. First, we describe current understanding of representative ion channels, namely TRPA1, TRPV1, and ANO1 (Part 1). Second, we discuss the multiple functions of TRP channels and ANO1 (Part 2). Finally, we propose the significance of those functions in clinical situations, including headache, migraine, and fungus infection (Parts 3 and 4)."
Moreover, we changed the title to: “Emerging perspectives on pain management by modulation of TRP channels and ANO1”. We described each ion channel in the title because we believe that clear identification of ion channel names better reflects the topic of the main issue i.e., “Ion channels in nociception”.
Other concerns:
1. The text calls out a Figure 1, but the copy I received had no figure included. On a related note, the manuscript could use many more figures than just one, graphical representations of the pathways discussed would go a long way in helping the reader sift through the massive amount of information the authors discuss in this review. Even a figure depicting the different types of TRP channels and their endogenous and exogenous agonist/antagonists would be helpful for the reader.
We apologize for our mistake. We have included additional figures (Figure 1, 2 and 3) and a table (Table 1) according to the reviewers’ suggestions.
2. While the authors discuss TRPV1 and TRPA1 individually in detail, they give very little information about the well-known cross-sensitization that occurs between the two channels.
We thank the reviewer for this comment. Actually, the molecular mechanism is an important point to discuss regarding TRP-associated pain sensation. We have added sentences to describe the TRPV1–TMEM100–TRPA1 interaction (p.11, lines 288–301) as below. Due to this suggestion, we believe that the extent of our discussion has been improved.
" TRPV1 is co-expressed with TRPA1 in DRG neurons. Because TRPA1 activity is enhanced by intracellular calcium, it had been thought that calcium influx through TRPV1 activation could affect TRPA1 function. However, the TRPV1 entity reduces the probability of TRPA1 ion channel opening accelerated by mustard oil [92]. It appears that TRPA1-associated pain is normally reduced by TRPV1 expression, which may be prevented by TMEM100 [92]. TMEM100 is a small membrane protein, and its expression pattern highly overlaps with CGRP. Interestingly, TRPA1 almost co-localizes with TRPV1, TMEM100, and CGRP in DRG neurons (Figure 1). Together, TRPV1, TRPA1, and TMEM100 form a complex, and the interaction between TRPV1 and TRPA1 is suppressed by interposition of TMEM100. Furthermore, a mutant peptide of TMEM100 (T100-Mut) can permeate the plasma membrane and disturb correct binding of TMEM100, thereby inhibiting TRPA1-associated pain-related behavior. This may provide a novel strategy for reducing pain sensation."
3. The section on migraines includes a paragraph about TRPM8, which appears relatively out of the blue, i.e. there was no discussion of TRPM8 in the beginning of the paper where TRP channels and their functions are discussed.
We have included a sentence on TRPM8. Nonetheless, we believe that the explanation in the ‘Headache and migraine’ section is easier for the audience to understand. Therefore, we have included an additional paragraph in this section (p.18, lines 494–502) as below.
" TRPM8 is found on both Aδ and C fiber afferents, and is important for the activation of peripheral sensory neurons by cold temperature. It is activated at non-noxious cold temperatures (< 26 °C) and by compounds that produce a cooling sensation such as menthol or eucalyptol [147, 148]. While its role as a cold sensor has been firmly established, it is not the case regarding its role in pain sensation. It is still under debate whether TRPM8 reduces or exacerbates pain sensation, and the most recent view on the matter is that TRPM8-expressing afferent fibers have the ability to both produce and alleviate pain, and the outcome will be determined by context (see for review [149, 150]). As such, TRPM8 has begun to gather attention in the migraine field. "
4. There are a few places where the authors could improve their use of English grammar.
We have modified any English grammar mistakes
Reviewer 3 Report
Abstract: OK
Introduction:
. Line 44:proteins, particularly ion channels that versatilyfunction in a range of
. Line 59 temperature increaseschanges that reach, too
. Line 62-69: reference?
. Line 114:protein (AKAP) in DRG neurons [29-32] that are activated by GPCRs [29-32].
. Line 130:The crystal structure of fungal ANO washas beenrecently determined at high
. Line 152:we “first”? the first means your group is the first to investigate ANO1 and TRPV1, or you mean the first step for your experiment is to …… Please clarify the sentence.
. Line 131-133: Rewrite as following: Furthermore, the structure of mouse is a dimer containing ten transmembrane regions (TMs) in one subunit by cryo-electron microscopy [37, 38], and the calcium binding….
. Line 135 From the structure analysis: Structural analysis showed that the dynamic
. Line 136: was inferred to: may be
. Line 138: is also likely “to be” important
. Line 140: can be is possibly
. Line 142: a very confusing sentence. Please clarify the “similar to ……” you tried to compare is TRPV1 inducing pain or the mechanisms underlying the pain?
. Line 143: characteristic”s” couldmay explain why “the”
. Line 155 T16Ainh-A01 (unpublished ? or any reference?).
. also
.
. Line 159 second application (reference)
. Line 160 reference
. Line 162: after 1 hr “spontaneously?”
. 163: are inhibited by the activation of the TRPV1
. 164: confusing sentence, TRPV1 function is low? What does this mean? You mean the expression amount of the TRPV1 decrease, or by calcium imaging or patch clamp you detect the current from TRPV1 decrease?
. 166: please redefine the sentence a possible……?
. Line 175-178: grammar errors, and please move to 1-2-1
. Line 186: TRPA1 can beis activated
. Line 187:increases during agonist application: during the application of a given agonist, as seen ….
. Line 191: not promote, “enhance”
. Line 193: please delete sentence after TRPA1, and adding “and the effects are bimodal”
. Line 198: rewrite: Although human TRPA1 does not show cold response, however, it responds to cold at approximately 18 C if oxidization with dehydroxylation at proline 394 occurs [71].
.
Line 206: downstream? Please redefine
. Line 208: could be: is
. Line 209: further more: moreover
. Line 212: after NGF application: after the presence of NGF
. Line 215-216: confusing sentence
. Line 217: could dependrelies on the activationsof both TRPA1
. line 224: CFA 1 day treatment and partiral sciatic …
. Line224-230: syntax errors
. Line 233-234: These pro-inflammatory components possibly contain likely includelipopolysaccharide (LPS) from Gram-negative bacteria that, which can activate
. Line 240: Meanwhile, these results suggest that LPS directly activates TRPA1. Indeed, LPS increases single channel activities in TRPA1-expressing CHO cells, and a novel relationships between bacteria and primary sensory nerves and that TRPA1 antagonists
. Line 251, syntax error
. Line 256: state? status
. Line 259: together? Collectively
. Line 276: damage from. Neuronal death may occur in the low level
. Line 280: could, Such acidification directly or indirectly induce…
. Line 282 can induce myelin damage: can damage myelin. Intracellular alkalization also affects TRPA1 activity [60].
.
. Line 284: therapeutics that enhancedTRPA1 inhibition may be beneficial to the treatment of the central nervous system diseases.
. Line 346: respectively, these symptoms are diverse and mayincluding hypersensitivity to light, sound and smell, fatigue, neck stiffness, yawning, mood change, nausea, vomiting, cutaneous allodynia and transient visual disturbances termed aura [106-109].
. Line 351: This: The
. Line 364: concerning: regarding
Line 367: The first hint thatTRPV1 is involved in migraine was the finding that it isco-expressed with CGRP in rat TG neurons [117, 118] and also in mouse dural afferent neurons [119], suggesting a crucial role for TRPV1 in the migraine.
Line 387: neurons may contribute to the direct sensitization of TRPV1,
. Line 390:including NGF [125], bradykinin and prostaglandin E2 are also inflammatory mediators released during neurogenic inflammation and are commonly used in an animal model of headache to induce a chronic state of trigeminal hypersensitivity[126-128].
. Line 395: activation: firing
. Line 400: status
Line 419: Obesity may increase the frequencies and intensities of migraine. A study showed that when mice were fed with …
. Line 439:A link with migraine could then be readily deduced from the ability of these compounds to generate headache, and TRPA1 activation in
. Line 441:the nasal mucosa was suspected to trigger headache whenirritants were inhaled.
. Line 463: creating: leading to
. Line 464: Based on these findings, the authors proposed
. Line 466:RNS can also act as TRPA1 agonists [25] that have been linked tomay cause headaches and
. Line 469:formedgenerated by a redox reaction between NO and H2S, could activate TRPA1 activationin the
. Line 471-472:ultimately resulted in vasodilation, andincreased meningeal blood flow, and could also account forthe headache phase of a migraine attack.
. Line 479: twins (reference)
. Line 481: found: identified
. Line 485:TRPM8 SNP variants, which were heterologously expressed in HEK293 cells, showed diverse channel activities by using calcium imaging experiments [158].
. Line 503: The contraryresults obtainedin these two studies may contribute to the different experimental settings
. Line 532: have limitedimpede the clinicaluse
. Line 534: initial clinical randomizedtrial
. Line 535: attacks; in this randomized study, and
. Line 536: AlthoughEven thoughthis outcome does not exclude the role contributionof TRPV1 toinmigraine pathology, it could indicatehighlights that targeting only TRPV1 selectively targeting TRPV1 aloneis a drop in the bucket not sufficient for acute treatment of migraine attack.
. Conclusion: weak, you need a solid and strong end for your review.
Author Response
Comments and Suggestions for Authors
Abstract: OK
Introduction:
. Line 44:proteins, particularly ion channels that versatilyfunction in a range of
. Line 59 temperature increaseschanges that reach, too
. Line 62-69: reference?
. Line 114:protein (AKAP) in DRG neurons [29-32] that are activated by GPCRs [29-32].
. Line 130:The crystal structure of fungal ANO washas beenrecently determined at high
. Line 152:we “first”? the first means your group is the first to investigate ANO1 and TRPV1, or you mean the first step for your experiment is to …… Please clarify the sentence.
. Line 131-133: Rewrite as following: Furthermore, the structure of mouse is a dimer containing ten transmembrane regions (TMs) in one subunit by cryo-electron microscopy [37, 38], and the calcium binding….
. Line 135 From the structure analysis: Structural analysis showed that the dynamic
. Line 136: was inferred to: may be
. Line 138: is also likely “to be” important
. Line 140: can be is possibly
. Line 142: a very confusing sentence. Please clarify the “similar to ……” you tried to compare is TRPV1 inducing pain or the mechanisms underlying the pain?
. Line 143: characteristic”s” couldmay explain why “the”
. Line 155 T16Ainh-A01 (unpublished ? or any reference?).
. also
.
. Line 159 second application (reference)
. Line 160 reference
. Line 162: after 1 hr “spontaneously?”
. 163: are inhibited by the activation of the TRPV1
. 164: confusing sentence, TRPV1 function is low? What does this mean? You mean the expression amount of the TRPV1 decrease, or by calcium imaging or patch clamp you detect the current from TRPV1 decrease?
. 166: please redefine the sentence a possible……?
. Line 175-178: grammar errors, and please move to 1-2-1
. Line 186: TRPA1 can beis activated
. Line 187:increases during agonist application: during the application of a given agonist, as seen ….
. Line 191: not promote, “enhance”
. Line 193: please delete sentence after TRPA1, and adding “and the effects are bimodal”
. Line 198: rewrite: Although human TRPA1 does not show cold response, however, it responds to cold at approximately 18 C if oxidization with dehydroxylation at proline 394 occurs [71].
. Line 206: downstream? Please redefine
. Line 208: could be: is
. Line 209: further more: moreover
. Line 212: after NGF application: after the presence of NGF
. Line 215-216: confusing sentence
. Line 217: could dependrelies on the activationsof both TRPA1
. Line 224: CFA 1 day treatment and partiral sciatic …
. Line224-230: syntax errors
. Line 233-234: These pro-inflammatory components possibly contain likely includelipopolysaccharide (LPS) from Gram-negative bacteria that, which can activate
. Line 240: Meanwhile, these results
suggest that LPS directly activates TRPA1.
Indeed, LPS increases single channel activities in TRPA1-expressing CHO cells, and a novel relationships between bacteria and primary sensory nerves and that TRPA1 antagonists
. Line 251, syntax error
. Line 256: state? status
. Line 259: together? Collectively
. Line 276: damage from. Neuronal death may occur in the low level
. Line 280: could, Such acidification directly or indirectly induce…
. Line 282 can induce myelin damage: can damage myelin. Intracellular alkalization also affects TRPA1 activity [60].
. Line 284: therapeutics that enhancedTRPA1 inhibition may be beneficial to the treatment of the central nervous system diseases.
. Line 346: respectively, these symptoms are diverse and mayincluding hypersensitivity to light, sound and smell, fatigue, neck stiffness, yawning, mood change, nausea, vomiting, cutaneous allodynia and transient visual disturbances termed aura [106-109].
. Line 351: This: The
. Line 364: concerning: regarding
. Line 367: The first hint thatTRPV1 is involved in migraine was the finding that it isco-expressed with CGRP in rat TG neurons [117, 118] and also in mouse dural afferent neurons [119], suggesting a crucial role for TRPV1 in the migraine.
. Line 387: neurons may contribute to the direct sensitization of TRPV1,
. Line 390:including NGF [125],
bradykinin and prostaglandin E2 are also inflammatory mediators released during neurogenic
inflammation and are commonly used in an animal model of headache to induce a chronic
state of trigeminal hypersensitivity[126-128].
. Line 395: activation: firing
. Line 400: status
. Line 419: Obesity may increase the frequencies and intensities of migraine. A study showed that when mice were fed with …
. Line 439:A link with migraine could then be readily deduced from the ability of these compounds to generate headache, and TRPA1 activation in
. Line 441:the nasal mucosa was suspected to trigger headache whenirritants were inhaled.
. Line 463: creating: leading to
. Line 464: Based on these findings, the authors proposed
. Line 466:RNS can also act as TRPA1 agonists [25] that have been linked tomay cause headaches and
. Line 469:formedgenerated by a redox reaction between NO and H2S, could activate TRPA1 activationin the
. Line 471-472:ultimately resulted in vasodilation, andincreased meningeal blood flow, and could also account forthe headache phase of a migraine attack.
. Line 479: twins (reference)
. Line 481: found: identified
. Line 485:TRPM8 SNP variants, which were heterologously expressed in HEK293 cells, showed diverse channel activities by using calcium imaging experiments [158].
. Line 503: The contraryresults obtainedin these two studies may contribute to the different experimental settings
. Line 532: have limitedimpede the clinicaluse
. Line 534: initial clinical randomizedtrial
. Line 535: attacks; in this randomized study, and
. Line 536: AlthoughEven thoughthis outcome does not exclude the role
contributionof TRPV1 toinmigraine pathology, it could indicatehighlights that targeting only TRPV1 selectively targeting TRPV1 aloneis a drop in the bucket not sufficient for acute treatment of migraine attack.
We thank the reviewer for their thorough checking of English grammar. We have modified the manuscript according to your suggestions.
. Conclusion: weak, you need a solid and strong end for your review.
We have changed the entire conclusion as below. We believe that our revised conclusion is more compelling.
" Studies from the last two decades show the importance of TRP channels in pain sensations caused by noxious temperatures and many chemicals. In particular, TRPA1 is activated in many conditions and its activity evokes an extremely uncomfortable sensation. Therefore, TRPA1 may be a crucial target for pain treatment, although TRPV1 contribution to more specific nociception cannot be disregarded. While some pharmaceutical companies are already focused on the development of TRPA1 and TRPV1 antagonists [189], ANO1 inhibition may also be effective in treating pain because the role of ANO1 is akin to an amplifier, and its suppression does not generate a painless condition. Namely, a level of nociception that is sufficient enough to sense damage for survival can be maintained in an ANO1-blocked state, but not with shutdown of the detectors, i.e., TRP channels.
Sensory systems do not only detect noxious stimuli but also participate in establishment of inflammation or chronic pain. Neural excitation induces CGRP release from nociceptor termini, inducing inflammation that can ultimately lead to sensitization of nociceptors. Therefore, inhibition of CGRP release by suppression of TRP channel activity is expected to provide relief for intractable pain, headache, and migraine. However, this strategy may result in dangerous secondary effects in some diseases, including fungus infection. Bone is often disrupted in Candida infection, a situation induced by osteoclast activity. It remains unclear why osteoclast activity is up-regulated during infection, but in this case, CGRP release from nociceptors becomes beneficial by suppressing two pathways: TNF-a release from myeloid cells and overactivation of osteoclasts. Thus, inflammation induced by CGRP is not detrimental in some pathological conditions, yet targeting it for pain relief might not always be the best strategy.
In fact, pain sensation negatively controls our physiological conditions, including our emotions. Recent reports indicate that complete abolishment of pain induces a severe pathological condition to our body. Therefore, we believe that the important point of pain management is to decrease pain to a tolerable level, but one that is sufficient enough to maintain natural protection against tissue damage. To develop this strategy, there is a need for the discovery of new TRP channels and ANO1 inhibitors that could be used concomitantly and adjusted depending on each patient’s condition."
Reviewer 4 Report
This review gives an introduction about the functions of some TRP channels and ANO1 in pain sensation. However, I am afraid that I cannot give favorable opinions on this review.
First, the three parts (sensory neurons, migraine, immunity) of the review does not seem well organized. There are no comparisons among them. They are in the same article probably only because they involve "TRP channels and ANO" and they are all "pains". The way that the authors organize the three parts is not friendly for readers to get general ideas of these channels' functions in three types of "pains".
Second, the channels the authors pick are very random. They pick "some of the TRP channels" and one "ANO1" channel, which is very partial. For example, it is reported that TRPA1, TRPV1 and TRPM3 work together for noxious heat sensing (Vandwauw I, et al, 2018 Nature), but TRPM3 (also a TRP channel) is completely not mentioned in this article. While ANO1 is not a TRP channel, I believe it is introduced only because the authors want to add "ANO1-TRP channel complex" (a paper from their own lab). A review is supposed to have a panoramic opinion on the issue, but the authors only discuss the parts that they are interested in, while ignoring other necessary components.
Third, the information in the review is biased. It seems to me that they spend "too much content" on the work from their own labs. In the ANO1-TRP complex part, they even analyze the data in details, which should not be in a review. Before they introduce the channels, they should at least let the readers know what neuron types the channels are expressed in, but they only generally say "DRG neurons" or "TG neurons", which is not informative.
To sum up, I believe the review is just an accumulation of materials. I doubt whether this review will be helpful to readers.
Author Response
This review gives an introduction about the functions of some TRP channels and ANO1 in pain sensation. However, I am afraid that I cannot give favorable opinions on this review.
First, the three parts (sensory neurons, migraine, immunity) of the review does not seem well organized. There are no comparisons among them. They are in the same article probably only because they involve "TRP channels and ANO" and they are all "pains". The way that the authors organize the three parts is not friendly for readers to get general ideas of these channels' functions in three types of "pains".
We have modified the presentation of our review. Further, we have deleted certain parts because the previous version actually focused on only TRP–ANO1 interactions. On reflection, we felt that overall we had covered another field. Therefore, we have added additional sentences to discuss extensively, for example, the TRPV1–TMEM100–TRPA1 interaction in CGRP-positive DRG neurons.
Second, the channels the authors pick are very random. They pick "some of the TRP channels" and one "ANO1" channel, which is very partial. For example, it is reported that TRPA1, TRPV1 and TRPM3 work together for noxious heat sensing (Vandwauw I, et al, 2018 Nature), but TRPM3 (also a TRP channel) is completely not mentioned in this article.
As the reviewer has noted, the finding in the Nature paper is a recent, significant discovery. According to their previous report in Neuron, TRPM3 is involved in heat sensation through excitation of primary sensory neurons. Therefore, we have added another paragraph on TRPV1–TRPA1–TRPM3 conjugation (p.12, line 331–p.13, line 345) as below and illustrated this in Figure 1.
" TRPM3 is a heat sensitive TRP channel that functionally couples with TRPV1 and TRPA1 [98]. Although TRPM3-deficient mice show a delayed tail flick at 57 °C, the effect of TRPM3 alone on heat sensation is unclear because tail flick behavior induced at 57 °C in TRPM3/TRPA1 double-deficient mice is no different to wild-type mice [4]. However, triple conjugation of TRPV1, TRPA1, and TRPM3 is important for detecting the noxious heat environment [4]. Withdrawal latency of TRPM3-deficient mice in the hot-plate test (50 °C) is the same as in wild-type mice [98]. Interestingly, this behavior disappears in TRPV1/TRPA1/TRPM3 triple-deficient mice, while the other responses to nociceptive stimuli are normal. Furthermore, wild-type and triple-deficient mice show a similar distribution on a gradient temperature plate (from 5 to 50 °C). In addition, CGRP-expressing DRG neurons are involved in heat sensation [62], and CGRP release from skin preparations is enhanced by the TRPM3 agonist, CIM0216, which is the same as for capsaicin treatment [99]. These findings indicate that the likely multiple function of TRPV1, TRPA1, and TRPM3 in peptidergic DRG neurons is to escape from a noxious heat environment."
While ANO1 is not a TRP channel, I believe it is introduced only because the authors want to add "ANO1-TRP channel complex" (a paper from their own lab). A review is supposed to have a panoramic opinion on the issue, but the authors only discuss the parts that they are interested in, while ignoring other necessary components.
Third, the information in the review is biased. It seems to me that they spend "too much content" on the work from their own labs. In the ANO1-TRP complex part, they even analyze the data in details, which should not be in a review. Before they introduce the channels, they should at least let the readers know what neuron types the channels are expressed in, but they only generally say "DRG neurons" or "TG neurons", which is not informative.
We agree with the reviewer’s suggestion. Actually, the previous version was unfair in another field. In the revised version, we have deleted all descriptions on the TRP–ANO1 interaction, except within the pain sensation description in the Introduction. Additionally, we have discussed the population of nociceptors with each channel in Figure 1. Further, we have added sentences (p.3, lines 57-60) in the Introduction describing basic understanding of primary sensory neurons as below.
" There are three types of nerves in primary sensory neurons, including Ab-, Ad-, and C-fibers. Ab-fibers are myelinated afferent nerves that respond to innocuous mechanical stimuli. Ad-fibers are also myelinated nerves, but alternatively this nervous pathway responds to rapid noxious stimuli. C-fibers are nonmyelinated nerves involved in slow pain [1]."
To sum up, I believe the review is just an accumulation of materials. I doubt whether this review will be helpful to readers.
We thank the reviewer for their comment. We have modified our manuscript, and added more discussion of recent reports, in particular on the TRPV1–TMEM100–TRPA1 interaction and TRPV1–TRPA1–TRPM3 conjugation. These additions have developed our discussion in this review. We believe that our manuscript is improved compared with our previous version.
Round 2
Reviewer 2 Report
The authors have addressed my earlier concerns.
Reviewer 3 Report
Accepted
Reviewer 4 Report
N/A